# Ureteral Stent and Percutaneous Nephrostomy in Managing Malignant Ureteric Obstruction of Gastrointestinal Origin: A 10 Years' Experience

**Elisa De Lorenzis** [1,2,*], **Elena Lievore** [1], **Matteo Turetti** [1], **Andrea Gallioli** [1], **Barbara Galassi** [3], **Luca Boeri** [1] **and Emanuele Montanari** [1,2]

1    Urology Unit, Fondazione IRCCS Ca' Granda Ospedale Maggiore Policlinico, 20122 Milan, Italy; elena.lievore01@gmail.com (E.L.); matteo.turetti@gmail.com (M.T.); andrea.gallioli@gmail.com (A.G.); dr.lucaboeri@gmail.com (L.B.); emanuele.montanari@unimi.it (E.M.)
2    Department of Clinical Sciences and Community Health, University of Milan, 20122 Milan, Italy
3    Division of Medical Oncology, Fondazione IRCCS Ca' Granda Ospedale Maggiore Policlinico, 20122 Milan, Italy; barbara.galassi@policlinico.mi.it
*    Correspondence: elisa.delorenzis@gmail.com; Tel.: +39-02-55034546; Fax: +39-02-5032-0584

**Abstract:** Background: Malignant ureteral obstruction (MUO) is variable in presentation and there is no consensus on its management, especially when caused by gastrointestinal (GI) malignancies. Our aim was to describe our experience with this oncological complication. Methods: We retrospectively analyzed the outcomes of ureteral stent and nephrostomy tube (NT) positioning for GI-related MUO from 2010 to 2020. We performed descriptive analysis, survival analysis, and uni- and multi-variate analysis. Results: We included 51 patients. NT was mainly used when bladder involvement occurred and when MUO revealed an ex novo cancer diagnosis. Survival was poorer in patients with new diagnoses and in those receiving no treatment after decompression. Moreover, MUO caused by upper-GI tumors was related to shorter overall survival. Conclusions: GI tumors causing MUO should be considered of poor prognosis. Treatment decisions should be weighted accurately by both specialists and the patient.

**Keywords:** ureteral obstruction; gastrointestinal cancer; percutaneous nephrostomy; decompression

## 1. Introduction

Malignant ureteral obstruction (MUO) can be caused by extrinsic compression of a tumor (primary neoplasm or relapse) and lymphadenopathy or by direct infiltration and it is a common sign in patients with advanced diseases. Moreover, in some cases, ureteral involvement can be a direct consequence of the surgical or radiotherapic treatment of the neoplasm itself.

The clinical presentation of MUO is variable and heterogenous, spanning from anuria and acute renal insufficiency to colic pain and urosepsis. In addition, it can be asymptomatic and discovered only by imaging during staging or follow-up. An MUO must be also suspected and investigated in the case of unknown or ex novo occurrence of renal impairment. Therefore, an MUO can not only affect patient's quality of life but also delay or contraindicate systemic therapy.

The primary aims of urinary diversion are symptoms relief and renal function improvement, which in turn may permit the start or the continuation of therapy, potentially prolonging patient's survival.

On the other hand, the insertion of a ureteral stent (US) or a nephrostomy tube (NT) is not devoid of complications (infection, urinary tract symptoms, bleeding, device encrustation or dislocation, and sometimes device failure, in the case of US) [1,2].

Due to the proximity to ureters, bladder, and urethra, gastrointestinal (GI) cancers, encompassing tumors arising from the proximal esophagus to the distal rectum, are, along with gynecological tumors, the most common extra-urological malignancies causing MUO. Despite advances in surgical techniques, chemotherapy, and radiation therapy, these cancers continue to take a tremendous toll on society, with numerous patients suffering from disease recurrence and progression. Furthermore, the clinical management of GI tumors could involve the urogenital tract due both to iatrogenic urologic injuries, occurring during abdominal and pelvic surgery [3], and to ureteral obstruction caused by the tumor itself.

To date, the optimal management of MUO is not clear and standardized. Some studies analyzed outcomes after NT or US insertion in patients affected by different types of malignancies [4–6], both urological and non-urological, but the superiority of one type of urinary diversion has not been demonstrated.

We sought to describe a population including only patients affected by gastrointestinal primary malignancies who had ureteral stents or nephrostomies inserted for MUO. To this aim, we analyzed:

1.  Outcomes data regarding renal function, hospitalization, complications, stent failure, and overall survival (OS) following insertion of US and NT
2.  Differences between patients treated with NT and those treated with US
3.  The rate and the characteristics of patients who received further oncological treatment after stent/nephrostomy insertion
4.  Potential risk factors associated with worse outcomes.

## 2. Results

A total of 51 patients were included in the study. Twenty-seven patients underwent US (53%) and 24 NT insertion (47%).

Table 1 lists the characteristics of the studied population.

In 14 patients (27.5%), the evidence of MUO and malignancy diagnosis were concurrent.

Regarding the clinical presentation, 13 patients (25.5%) had lumbar pain, 7 presented with anuria (13.7%), 8 with fever (15.7%), and 23 were asymptomatic (45.1%), and the diagnosis of MUO was made through imaging alone. Among the asymptomatic patients, 10 (43.5%) had severe hydronephrosis. Out of seven patients with anuria, hydronephrosis was severe in three cases, moderate in additional three, and mild in one case. Six of these patients (85.7%) had bilateral hydronephrosis. Two patients (3.9%) needed a session of hemodialysis before the urological procedure.

The median time between malignancy diagnosis and urinary diversion was 8 (IQR 0–30.75) months.

Before urinary diversion, 13 patients (25.5%) underwent only chemotherapy, 2 (3.9%) only radiotherapy, and 13 (25.5%) chemoradiotherapy.

In 10 patients (19.6%), stent insertion was unsuccessful, and consequently an NT was positioned.

Table 2 shows peri-operative variables of the study population. The median length of hospital stay was 9 days (IQR 3-17). Complications occurred in seven patients (13.7%), and eight patients had subsequent US failure (29.6%).

Comparing patients with stent failure and those with functioning stent, no differences were found, except for the type of primary malignancy (*p* = 0.02; all stent failures occurred in rectal tumors).

The overall median survival from urinary diversion was 10.5 months (IQR 4–17.25), and two patients (3.9%) died during hospital stay.

Table 3 compares patients who underwent US positioning versus those with NT positioning. The two groups resulted homogeneous and differed significantly only for the presence of bladder invasion at cystoscopy, which was more frequently present in the NT group, and for the period of time between malignancy diagnosis and urinary diversion, which was significantly shorter in the NT group (all *p* < 0.05).

**Table 1.** Patients' clinical characteristics (medians and interquartile ranges (IQR) for continuous variables and frequencies for categorical variables).

| Parameter | Overall (*n* = 51) |
|---|---|
| Age at surgery (years) | 70 (58–76) |
| Male n | 20 (39.2%) |
| Female n | 31 (60.8%) |
| ECOG [#] Performance status n (%) | |
| 0–1 | 37 (72.5%) |
| 2–3 | 14 (27.5%) |
| Type of malignancy n (%) | |
| Rectal | 14 (27.5%) |
| Colon | 28 (54.9%) |
| Gastric | 5 (9.8%) |
| Pancreatic | 3 (5.9%) |
| Appendicular | 1 (1.9%) |
| Location of obstruction | |
| Upper | 3 (5.9%) |
| Middle | 10 (19.6%) |
| Lower | 38 (74.5%) |
| Side | |
| Right | 14 (27.4%) |
| Left | 16 (31.4%) |
| Bilateral | 21 (41.2%) |
| Cause of obstruction | |
| Unknown | 7 (13.7%) |
| Mass | 21 (41.2%) |
| Lymphadenopathy | 4 (7.8%) |
| Ureteral infiltration | 9 (17.7%) |
| Carcinomatosis | 10 (19.6%) |
| Hydronephrosis degree | |
| Mild | 7 (13.7%) |
| Moderate | 23 (45.1%) |
| Severe | 21 (41.2%) |
| Bladder invasion | |
| Yes | 7 (13.7%) |

[#] ECOG = Eastern Cooperative Oncology Group.

**Table 2.** Peri-operative variables (medians and IQR for continuous variables and frequencies for categorical variables).

| Parameter | Overall (*n* = 51) |
|---|---|
| Type of urinary diversion | |
| Ureteral stent | 27 (53%) |
| Nephrostomy tube | 24 (47%) |
| Hospital stay (days) | 9 (3–17) |
| Stent failure n | 8 (15.7%) |
| Time to stent failure (days) | 30 (23.5–128.5) |
| Complications n (%) | |
| No | 44 (86.3%) |
| Yes | 7 (13.7%) |
| Type of complication n (%) (*n* = 7) | |
| Fever | 2 (28.6%) |
| Pyelonephritis | 1 (14.2%) |
| Encrustation | 2 (28.6%) |
| Sepsis | 2 (28.6%) |
| Deaths n (%) | 43 (84.3%) |
| Overall survival (months) | 10.5 (4–17.2) |

**Table 3.** Comparison between patients treated with ureteral stent (US) and those with nephrostomy tube (NT).

| Parameter | US Group (*n* = 27) | NT Group (*n* = 24) | *p* |
|---|---|---|---|
| Age at surgery (years) | 70 (60–77) | 68 (57–73.7) | 0.44 |
| Gender | | | |
| Female | 16 (59.2%) | 15 (62.5%) | 0.8 |
| Male | 11 (40.8%) | 9 (37.5%) | |
| ECOG Performance status n (%) | | | |
| 0–1 | 20 (74%) | 17 (70.8%) | 0.8 |
| 2–3 | 7 (26%) | 7 (29.2%) | |
| Type of malignancy n (%) * | | | |
| Upper GI tract | 5 (18.5%) | 4 (16.7%) | 0.8 |
| Lower GI tract | 22 (81.5%) | 20 (83.3%) | |
| Location of obstruction | | | |
| Upper | 1 (3.7%) | 2 (8.3%) | |
| Middle | 7 (25.9%) | 3 (12.5%) | 0.4 |
| Lower | 19 (70.4%) | 19 (79.2%) | |
| Laterality | | | |
| Monolateral | 17 (63%) | 13 (54.2%) | 0.5 |
| Bilateral | 10 (37%) | 11 (45.8%) | |
| Cause of obstruction | | | |
| Unknown | 3 (11.1%) | 4 (16.7%) | |
| Mass | 11 (40.7%) | 10 (41.7%) | |
| Lymphadenopathy | 2 (7.4%) | 2 (8.3%) | 0.8 |
| Ureteral infiltration | 4 (14.9%) | 5 (20.8%) | |
| Carcinomatosis | 7 (25.9%) | 3 (12.5%) | |
| Hydronephrosis degree | | | |
| Mild | 2 (7.4%) | 5 (20.8%) | |
| Moderate | 14 (51.9%) | 9 (37.5%) | 0.3 |
| Severe | 11 (40.7%) | 10 (41.7%) | |
| Bladder invasion | | | |
| Yes | 1 (3.7%) | 6 (25%) | **0.03** |
| No | 26 (96.3%) | 18 (75%) | |
| Time between malignancy diagnosis and urinary diversion (months) | 18 (7–36) | 1 (0–12) | **0.005** |
| Previous chemotherapy | | | |
| Yes | 17 (63%) | 14 (58.3%) | 0.7 |
| No | 10 (37%) | 10 (41.7%) | |
| Previous radiotherapy | | | |
| Yes | 6 (22.2%) | 9 (37.5%) | 0.2 |
| No | 21 (77.8%) | 15 (62.5%) | |
| Preoperative creatinine (mg/dL) | 1.5 (1–3.7) | 1.4 (0.9–2) | 0.6 |
| Preoperative eGFR (mL/min/1.73 m$^2$) | 37 (13.2–71.5) | 41.5 (24.2–71.5) | 0.5 |
| Hospital stay (days) | 5 (2–13) | 12 (4.2–21.7) | **0.04** |
| Postoperative creatinine (mg/dL) | 1 (0.6–1.5) | 0.9 (0.7–1.2) | 0.8 |
| Postoperative eGFR (mL/min/1.73 m$^2$) | 68 (40–92.5) | 72 (45.2–88.2) | 0.6 |
| Complications | | | |
| Yes | 6 (22.2%) | 1 (4.2%) | 0.06 |
| No | 21 (77.8%) | 23 (95.8%) | |
| Postoperative oncological treatment | | | |
| Yes | 16 (59.3%) | 9 (37.5%) | 0.1 |
| No | 11 (40.7%) | 15 (62.5%) | |
| Overall survival (months) | 11 (6.7–19) | 8 (3–16.5) | 0.2 |

GI = gastrointestinal. * Upper GI malignancies include gastric, pancreatic, and appendicular malignancies; lower GI malignancies include rectal and colon malignancies; eGFR = estimated glomerular filtration rate. Mann–Whitney U test was used for continuous variables, and Chi-square test for categorical variables.

Table 4 compares patients receiving or not further treatment after MUO management. Performance status, time between cancer and MUO diagnosis, and survival time were all significantly ($p < 0.05$) related to post-urinary diversion treatment decision.

**Table 4.** Comparison between patients who underwent oncological therapy after urinary diversion with those not receiving therapy.

| Parameter | Without Therapy (*n* = 26) | With Therapy (*n* = 25) | *p* |
|---|---|---|---|
| Age at surgery (years) | 65.5 (58–74.5) | 71 (57–76) | 0.8 |
| Gender | | | |
| Female | 19 (73.1%) | 12 (48%) | 0.06 |
| Male | 7 (26.9%) | 13 (52%) | |
| ECOG Performance status n (%) | | | |
| 0–1 | 15 (57.7%) | 22 (88%) | **0.015** |
| 2–3 | 11 (42.3%) | 3 (12%) | |
| Type of malignancy n (%) * | | | |
| Upper GI tract | 6 (23.1%) | 3 (12%) | 0.3 |
| Lower GI tract | 20 (76.9%) | 22 (88%) | |
| Location of obstruction | | | |
| Upper | 1 (3.8%) | 2 (8%) | 0.7 |
| Middle | 6 (23.1%) | 4 (16%) | |
| Lower | 19 (73.1%) | 19 (76%) | |
| Laterality | | | |
| Monolateral | 14 (53.8%) | 16 (64%) | 0.5 |
| Bilateral | 12 (46.2%) | 9 (36%) | |
| Cause of obstruction | | | |
| Unknown | 4 (15.4%) | 3 (12%) | |
| Mass | 8 (30.8%) | 13 (52%) | |
| Lymphadenopathy | 1 (3.8%) | 3 (12%) | 0.2 |
| Ureteral infiltration | 5 (19.2%) | 4 (16%) | |
| Carcinomatosis | 8 (30.8%) | 2 (8%) | |
| Hydronephrosis degree | | | |
| Mild | 4 (15.4%) | 3 (12%) | |
| Moderate | 13 (50%) | 10 (40%) | 0.6 |
| Severe | 9 (34.6%) | 12 (48%) | |
| Bladder invasion | | | |
| Yes | 5 (19.2%) | 2 (8%) | 0.2 |
| No | 21 (80.8%) | 23 (92%) | |
| Time between malignancy diagnosis and urinary diversion (months) | 3.5 (0–12.7) | 18.5 (1.25–37.5) | **0.007** |
| Previous chemotherapy | | | |
| Yes | 14 (53.8%) | 17 (68%) | 0.3 |
| No | 12 (46.2%) | 8 (32%) | |
| Previous radiotherapy | | | |
| Yes | 8 (30.8%) | 7 (28%) | 0.8 |
| No | 18 (69.2%) | 18 (72%) | |
| Preoperative creatinine (mg/dL) | 1.9 (1.1–3.7) | 1.4 (0.8–1.8) | 0.07 |
| Preoperative eGFR (mL/min/1.73 m$^2$) | 28 (12.5–60) | 43 (33.5–73.5) | 0.07 |
| Type of urinary diversion | | | |
| US | 11 (42.3%) | 16 (64%) | 0.1 |
| NT | 15 (57.7%) | 9 (36%) | |

**Table 4.** *Cont.*

| Parameter | Without Therapy (*n* = 26) | With Therapy (*n* = 25) | *p* |
|---|---|---|---|
| Stent failure | | | |
| Yes | 5 (19.2%) | 3 (12%) | 0.5 |
| No | 21 (80.8%) | 22 (88%) | |
| Hospital stay (days) | 13 (4.7–20.2) | 4 (2–11.5) | 0.06 |
| Postoperative creatinine (mg/dL) | 1 (0.7–1.4) | 0.9 (0.6–1.4) | 0.4 |
| Postoperative eGFR (mL/min/1.73 m$^2$) | 68 (40–86.5) | 75 (41.7–95) | 0.3 |
| Complications | | | |
| Yes | 3 (11.5%) | 4 (16%) | 0.6 |
| no | 23 (88.5%) | 21 (84%) | |
| Overall survival (months) | 5 (2–14.2) | 15 (8.2–29.5) | **0.003** |

GI = Gastrointestinal; * Upper GI malignancies include gastric, pancreatic, and appendicular malignancies; lower GI malignancies include rectal and colon malignancies. Mann–Whitney U test was used for continuous variables, and Chi-square test for categorical variables.

Performance status and time between diagnosis and urinary diversion were predictors of no post-procedural oncological treatment (*p* = 0.022 95% CI 0.04–0.78 and *p* = 0.037 95% CI 1.002–1.07, respectively) in univariate analysis (Table 5).

**Table 5.** Univariable logistic regression analysis of variables for further oncological treatment after urinary diversion.

| Parameter | *P* Value | 95% CI |
|---|---|---|
| Age at surgery (years) | 0.96 | 0.95–1.05 |
| Gender | 0.07 | 0.89–0.96 |
| ECOG Performance status | **0.02** | 0.04–0.78 |
| Type of malignancy * Upper GI tract Lower GI tract | 0.3 | 0.48–9.98 |
| Location of obstruction | 0.9 | 0.37–2.49 |
| Laterality | 0.4 | 0.21–2.01 |
| Cause of obstruction | 0.1 | 0.47–1.07 |
| Hydronephrosis degree | 0.4 | 0.63–3.22 |
| Bladder invasion | 0.2 | 0.06–2.08 |
| Time between malignancy diagnosis and urinary diversion | **0.037** | 1.002–1.07 |
| Previous chemotherapy | 0.3 | 0.58–5.68 |
| Previous radiotherapy | 0.8 | 0.26–2.92 |
| Preoperative creatinine | 0.4 | 0.75–1.12 |
| Preoperative eGFR | 0.07 | 0.99–1.04 |
| Type of urinary diversion | 0.12 | 0.13–1.27 |
| Hospital stay (days) | 0.13 | 0.89–1.01 |
| Postoperative creatinine | 0.5 | 0.22–2.07 |
| Postoperative eGFR | 0.3 | 0.99–1.03 |
| Complications | 0.6 | 0.29–7.30 |

GI = Gastrointestinal; * Upper GI malignancies include gastric, pancreatic, and appendicular malignancies; lower GI malignancies include rectal and colon malignancies.

Multivariate analysis showed that a low performance status was the only independent predictor of further oncological treatment after NT and US positioning ($p = 0.03$ 95% CI 0.03–0.87), after adjusting for time between diagnosis and urinary diversion

An analysis of survival showed that patients affected by malignancies originated from the upper gastrointestinal tract had worse survival than those with lower-GI tumors (Figure 1).

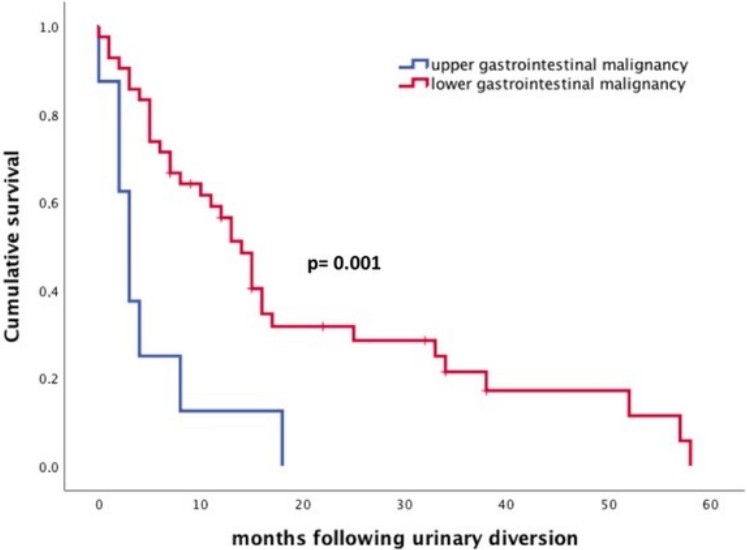

**Figure 1.** Kaplan–Meier curve for overall survival according to the type of malignancy.

As expected, patients who did not receive oncological therapy after urinary diversion had a significantly short survival time (Figure 2).

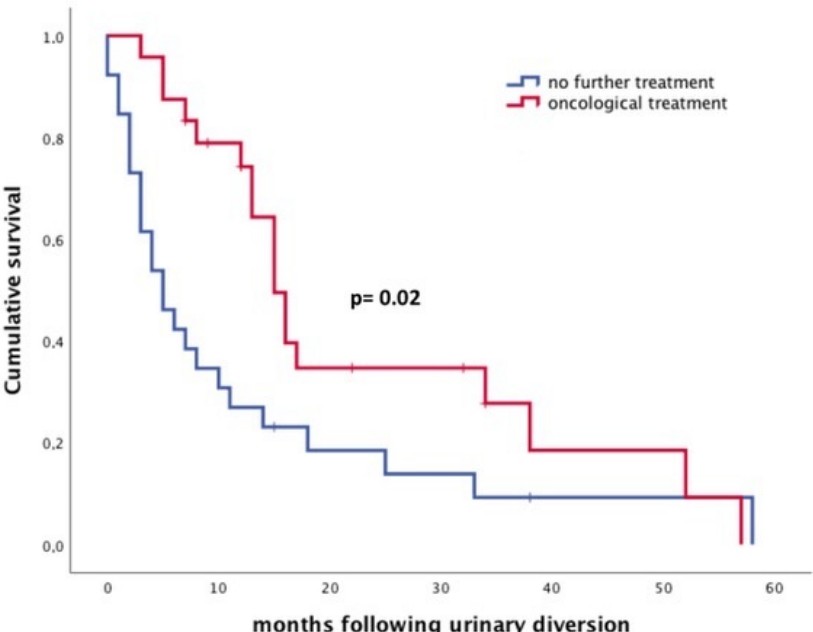

**Figure 2.** Kaplan–Meier curve for overall survival according to further oncological treatment after urinary diversion.

Univariate Cox regression demonstrated that upper gastrointestinal cancers, no further treatment, time between diagnosis and urinary diversion lower than 8 months from tumor diagnosis, and low performance status were associated with poor overall survival (Table 6).

**Table 6.** Univariate Cox regression analysis.

| Parameter | *p* Value | Hazard Ratio | 95% CI |
|---|---|---|---|
| ECOG Performance status | 0.000 | 4.5 | 2.2–9.1 |
| Type of malignancy (upper/lower GI tract) | 0.003 | 3.4 | 1.5–7.5 |
| Postoperative oncological treatment | 0.026 | 0.49 | 0.26–0.92 |
| Time between diagnosis and urinary diversion (</> 8 months) | 0.048 | 0.51 | 0.27–0.99 |

At multivariate Cox regression, only performance status was associated with survival ($p = 0.000$ 95% Hazard ratio 4.34 CI 2.11–8.9) after accounting for type of malignancy, further oncological treatment, and time between diagnosis and urinary diversion.

## 3. Discussion

There is no consensus regarding the optimal choice between ureteral stent or nephrostomy tube placement in case of GI-related MUO. European Association of Urology guidelines on pain management [7] suggest active treatment of symptomatic hydronephrosis and drainage of only one kidney in asymptomatic patients, although no strong evidence is provided to support this therapeutic decision [8]. No study has thoroughly explored which patients benefit more from such intervention in term of outcomes and quality of life. For example, a recent study conducted in the United States revealed that retrograde ureteral stenting was preferred for female patients and for those with lymphoma, ovarian, or colon cancer [9]. Surveys have shown no agreement for preferred management, i.e., US and nephrostomy, among urologists and oncologists [10,11].

This is, to our knowledge, the first study to address specifically the topic of gastrointestinal malignancies causing MUO. The choice between NT and US placement in these tumors may be driven by specific tumor features with respect to other tumors.

According to some studies, MUO caused by gastrointestinal tumors is associated with a worse prognosis with respect to other malignancies [4,12] and with a greater likelihood of stent failure [13,14]. However, the rate of stent failure in the present study (14.8%) is low compared to the 31% rate reported in a recent meta-analysis of gastrointestinal cancer [8].

Previous studies have identified several factors associated with stent failure in patients with MUO, including the pre-stenting serum creatinine level, performance status, and degree of hydronephrosis [15]. In our study, stent failure occurred only in patients with rectal malignancies. Kamiyama et al. [13] reported that gastrointestinal tumors, in general, have a higher probability of stent failure, with no mention of the specific type of cancer. In addition, they reported a significant correlation between bladder invasion and US failure. On the contrary, in our study bladder invasion was related to the placement of an NT in first instance, mainly because of failure of endoscopic US placement, as reported also by Guachetà-Bomba et al. [16].

The analysis of patients treated with NT versus US showed that the time between the diagnosis of the GI malignancy and MUO was significantly shorter in patients treated with NT, with 14 cases in which NT positioning and tumor diagnosis were concomitant (ex novo diagnosis). This may be explained by the fact that GI malignancies presenting with MUO are usually in advanced clinical stage and, consequently, they probably cannot be subjected to US positioning. In addition, the length of hospital stay for NT patients was longer than that for US patients, even if fewer complications were reported in the former group. The longer hospitalization may be explained by the need of collateral diagnostic and therapeutic procedures in newly diagnosed malignancies. Studies regarding solely NT placement in MUO, in fact, confirmed that non-urological malignancies, and in particular gastrointestinal cancers, are associated with poorer prognosis [17]. Our overall rate of complications was lower when compared to that of 41% reported in a recent review [8]. Specifically, only one of seven common complications, i.e., pyelonephritis, occurred in the NT group.

Further oncological treatment after US or NT placement was performed in approximately half of the patients. This is in line with Folkard et al. [6] who reported that 30.4% of patients treated with NT underwent further chemotherapy. Of note, Jeong et al. [4] described a short median survival time even if chemotherapy was used and explained the finding by the fact that the majority of patients in their cohort had GI cancers, which are intrinsically associated with poor prognosis. In our cohort, the significant factors recommending therapy after NT or US were a good performance status and a longer time between cancer and MUO diagnosis, but only a good performance status was confirmed by multivariate analysis. As demonstrated by our analysis, the general conditions of the patient also affected the overall survival.

When focusing on the type of GI tumor treated, our study showed that the survival time shortened considerably when MUO was caused by upper gastrointestinal tumors, even if there was no difference in treatment decisions between the two groups. Our observation relates to the fact that lower-GI tumors, namely, rectal and colonic, usually cause MUO through direct pelvic invasion, while upper-GI tumors usually cause MUO via carcinomatosis or lymph node spread. This may also relate to reported shorter survival times for patients with upper-ureteral obstruction [4]. Pavlovic et al., in addition, described lower rates of stent failures in patients with lower-GI malignancies, specifically [18].

Considering survival after urinary diversion, all patients showed short overall survival, especially those who did not receive any treatment after MUO management. These findings are in line with several other studies [5,6] which reported a very poor prognosis when cancer patients were diagnosed with MUO. The literature reports the presence of a malignancy that is unresectable or unsuitable for chemotherapy [19] and gastric and pancreatic primary tumors [20] as significant predictors for shorter overall survival. On the other hand, the possibility of further treatment may be an indication for urinary decompression [21]. Overall, data indicate that in the presence of a MUO, especially for gastrointestinal malignancies, prognosis is often unfortunate. Therefore, decision on whether to treat and on treatment choice should be weighted not only from a clinical and biochemical perspective, but also on the basis of patient's preference, who may choose life quality rather than life lengthening.

Patients with an ex novo diagnosis presenting with MUO are usually at such advanced stages that no effective treatment is available. Consequently, the decision to proceed with an invasive urinary diversion must be carefully evaluated with the patient, who should be extensive counselled about life expectancy, potential complications of the procedure, and subsequent quality of life.

On the other hand, asymptomatic cases of MUO should always be discussed in multidisciplinary settings, especially when caused by GI tumors, since urological expertise in this field is obviously limited in terms of prognosis and potential treatment.

Our study is not devoid of limitations. First, it is a retrospective single-center study, so the number of included cases was limited. We did not have data on quality of life of the patients treated, which would provide useful information for best treatment decisions.

## 4. Materials and Methods

We retrospectively reviewed patients affected by gastrointestinal malignancies (confirmed by histology) who underwent US or NT tube insertion in our hospital from January 2010 to January 2020. The first step consisted in analyzing the surgical database using the procedure codes for "percutaneous nephrostomy without fragmentation" (55.03) and "ureteral catheterization" (59.8) based on International Classification of Diseases, Ninth Revision, Clinical Modification (ICD-9-CM) [22]. From this group, all patients with a diagnosis of colon/rectal/gastric or pancreatic malignancy and a radiological evidence by computerized tomography (CT) scan or magnetic resonance imaging (MRI) of ureteral obstruction were included. Patients with an iatrogenic ureteral injury or a ureteral obstruction due to benign disease (e.g., ureteral stones) and patients submitted to urinary diversion for a nonobstructive cause (e.g., fistula) were excluded from the study. Patients who underwent prophylactic stent [23] before surgery were not included in the analysis.

The choice of urinary diversion (US or NT) was based on patients' characteristics (fitness for anesthesia, performance status, and, obviously, consent to the procedure) and tumor factors (bladder/trigone involvement, extensive ureteral invasion, and degree of extrinsic obstruction).

Stent failure was defined as a worsening in renal functional and/or persistent or progressive hydronephrosis and/or recurrent episodes of acute flank pain after US placement. In patients with a single kidney or a bilateral obstruction, anuria was considered a sign of US failure.

Demographic, clinical, and follow-up data were collected. The performance status of each patient was evaluated in accordance with the Eastern Cooperative Oncology Group (ECOG) criteria [24]. Type of malignancy, time between tumor diagnosis and ureteral decompression, location (upper, middle, lower ureter) and cause of obstruction (primary or recurrent mass, lymphadenopathy, ureteral infiltration, and carcinomatosis), bladder invasion (diagnosed by CT scan and/or cystoscopy), previous chemotherapy and/or radiotherapy, clinical presentation (flank pain, fever, renal function impairment, or evidence of hydronephrosis during imaging), hydronephrosis degree (basing on CT, MRI, and pyelography), and preoperative creatinine levels and estimated glomerular filtration rate (eGFR) were analyzed. We estimated the eGFR according to the Chronic Kidney Disease-Epidemiology Collaboration (CKD-EPI) [25].

Studied postoperative items included creatinine, eGFR, and length of hospital stay. Postoperative complications and readmission within 90 days were recorded. The follow-up was scheduled according to oncological indications. We also evaluated further oncological and surgical treatment and survival time.

Data collection followed the principles outlined in the Declaration of Helsinki. All patients signed an informed consent, agreeing to the surgery and to share their own anonymous information for future studies. This study was conducted retrospectively, collecting data obtained for clinical purposes, and all the procedures were performed as part of routine care. Consequently, our study did not need ethical approval.

### 4.1. Surgical Technique

In all cases, the procedures were performed by urologists. Stent positioning and subsequent change every 6 months were performed retrogradely by cystoscopy under fluoroscopic guidance, with patients under monitored sedation and in a lithotomy position. Usually, the US used was a polyurethane phosphorylcholine-treated double-J 6-7-fr, 26–28 cm long (Yellow-star tumor stent, Urotech GmbH Rohrdorf OT, Achenmühle, Germany)

If US placement failed, patients underwent NT placement by ultrasound-guided percutaneous puncture using an 18-fr needle, followed by insertion of a guidewire. Usually, a poliurethane 8-fr, 30 cm-long pigtail catheter (Teleflex Medical, Athlone, Co Westmeath, Ireland) was positioned under radiological guidance with a final pyelography, using iodine contrast medium. NT were changed every 45 days under fluoroscopic guidance. NT positioning and change were performed under local anesthesia.

Before device replacement, urine culture was performed for all patients, who also underwent blood exams.

### 4.2. Statistical Analysis

Data are presented as medians (interquartile ranges, IQR) or frequencies (proportions). A 95% CI was estimated for the association of categorical parameters. The statistical significance of differences in medians and proportions were tested with the Mann–Whitney test and Chi-square test, as indicated. Descriptive statistics were used to assess potential differences in terms of clinical parameters and intraoperative and postoperative characteristics between the US and the NT groups and between treated and non-treated patients after urinary diversion.

The Kaplan–Meier method was used to assess survival, and the relation between clinical variables and survival time was analyzed using the log-rank test. Cox regression analysis tested the predictors of overall survival.

Univariable logistic regression analyses were performed to determine potential predictors of further oncological treatment after urinary diversion. Of these variables, only those that had a *p* value < 0.05 by univariate analysis were included in the multivariate logistic model. Statistical analyses were performed using SPSS v.26 (IBM Corp., Armonk, NY, USA). All tests were two-sided, and statistical significance was established at $p < 0.05$.

## 5. Conclusions

The choice of urinary diversion technique for patients with MUO caused by GI malignancies varies depending on patient's presentation. When cancer presents with MUO, it should be considered of poor prognosis, especially if originating from the upper GI tract. In addition, patients with a low performance status and no treatment option available after MUO management have the worst prognosis. The choice of treatment should be driven by a thorough multidisciplinary consult considering also the patient's preferences.

**Author Contributions:** Conceptualization, E.D.L.; formal analysis, E.D.L., A.G.; data collection, E.L., M.T., E.D.L.; writing—original draft preparation, E.D.L. and E.L.; writing—review and editing, E.D.L., E.L., A.G., L.B.; supervision, B.G., E.M. All authors have read and agreed to the published version of the manuscript.

**Funding:** This research received no external funding.

**Conflicts of Interest:** The authors declare no conflict of interest.

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
