# Peer review of "Ureteral Stent and Percutaneous Nephrostomy in Managing Malignant Ureteric Obstruction of Gastrointestinal Origin: A 10 Years’ Experience"

_gastrointestdisord, doi:10.3390/gidisord2040041_

Round 1

Reviewer 1 Report

This study provides an interesting retrospective analysis of patients with MUO due to GI cancer.

I only have a few minor issues to address:

Table 4 - patients ‘who’ underwent…

Table 5 on page 6 correct the p-values for the creatinine

Materials and methods

Line 208 - ‘in analyzing’

Author Response

We are very grateful to the Reviewer 1 for the insightful comments to our paper.

Here is the list of the responses to the comments and the changes made in the manuscript:

1. Table 4 - patients ‘who’ underwent…

Thank you for your comment, we corrected the text accordingly

2. Table 5 on page 6 correct the p-values for the creatinine

Thank you for the comment. We revised our analysis to check for the p values and we did not find any discrepancies.

Materials and methods

3. Line 208 - ‘in analyzing’

Thank you for your comment, we corrected the text accordingly

Reviewer 2 Report

De Lorenzis and coauthors presented an interesting study: Ureteral stent and percutaneous nephrostomy in managing malignant ureteric obstruction of gastrointestinal origin. A ten years’ experience.

The authors addressed an interested topic, presented a retrospective analysis, enrolled 51 patients, highlighted the strengths and limitations of their study, and presented some interesting results and conclusions.

Closer look at the results have raised some comments:

  1. The authors should briefly describe ECOG performance status.
  2. How many asymptomatic patients had severe hydronephrosis?
  3. What is the percentage of bilateral hydronephrosis?
  4. What degree of hydronephrosis did patients with anuria have? Did they have bilateral hydronephrosis?
  5. What was the indication for hemodialysis?
  6. What method was used for eGFR?
  7. Who performed the NT? Radiologist or urologist?

The authors should accept and discuss these comments.

Author Response

We are very grateful to the Reviewer 2 for the insightful comments to our paper.

Here is the list of the responses to the comments and the changes made in the manuscript:

1. The authors should briefly describe ECOG performance status.

Thank you for the observation. We added a brief description and a reference regarding the score (line 228)

2. How many asymptomatic patients had severe hydronephrosis?

Thank you for the comment. We added the information in Results section (line 73). There were 10 out of 23 asyptomatic patients with severe hydronephrosis.

3. What is the percentage of bilateral hydronephrosis?

Thank you for the observation. The percentage of bilateral hydronephrosis is 41,2% (21 patients) as reported in table 1.

4. What degree of hydronephrosis did patients with anuria have? Did they have bilateral hydronephrosis?

Thank you for the comment. Patients with anuria had bilateral hydronephrosis in 6 over 7 cases. Hydronephrosis was severe in 3, moderate in 3 and mild in 1 case.  We added the information on results section, line 74

5. What was the indication for hemodialysis?

Thank you for the observation. The need for hemodialysis was indicated by the nephrology specialist after a consultation. Most cases it was severe hyperkalemia and/or metabolic acidosis.

6. What method was used for eGFR?

Thank you for the observation. We used CKD EPI creatinine 2009 equation. We modified the text accordingly (line 236), and we added an explanatory reference.

7. Who performed the NT? Radiologist or urologist?

Thank you for the comment. In all cases the procedures were performed by urologists. The information is reported in materials and methods, paragraph on surgical technique (line 248)